# Prognostic Impact of AHNAK2 Expression in Patients Treated with Radical Cystectomy

**DOI:** 10.3390/cancers13081748

**Published:** 2021-04-09

**Authors:** Dai Koguchi, Kazumasa Matsumoto, Yuriko Shimizu, Momoko Kobayashi, Shuhei Hirano, Masaomi Ikeda, Yuichi Sato, Masatsugu Iwamura

**Affiliations:** Department of Urology, Kitasato University School of Medicine, 1-15-1 Kitasato Minami-ku Sagamihara, Kanagawa 252-0374, Japan; dai.k@med.kitasato-u.ac.jp (D.K.); yulico@med.kitasato-u.ac.jp (Y.S.); momoko_dus@yahoo.co.jp (M.K.); s.hirano@med.kitasato-u.ac.jp (S.H.); ikeda.masaomi@grape.plala.or.jp (M.I.); sato.yuichi@kobal.co.jp (Y.S.); miwamura@med.kitasato-u.ac.jp (M.I.)

**Keywords:** bladder cancer, radical cystectomy, AHNAK2, prognosis

## Abstract

**Simple Summary:**

Unfavorable results following radical cystectomy for bladder cancer (BCa) highlights a critical need for a novel prognostic molecular biomarker with potential therapeutic benefits. In the present study, the expression levels of AHNAK2 in specimens obtained by radical cystectomy were classified as “low expression” or “high expression” by immunohistochemical staining. Then, we retrospectively evaluated associations between the two AHNAK2 expression patterns and the prognoses in terms of recurrence-free survival (RFS) and cancer-specific survival (CSS). Our multivariate analysis, adjusting for the effects of clinicopathological features, showed that the high expression level of AHNAK2 was an independent risk factor for RFS and CSS. The present study showed that AHNAK2 acts as a novel prognostic biomarker in patients with radical cystectomy for BCa.

**Abstract:**

Data regarding expression levels of AHNAK2 in bladder cancer (BCa) have been very scarce. We retrospectively reviewed clinical data including clinicopathological features in 120 patients who underwent radical cystectomy (RC) for BCa. The expression levels of AHNAK2 in the specimens obtained by RC were classified as low expression (LE) or high expression (HE) by immunohistochemical staining. Statistical analyses were performed to compare associations between the two AHNAK2 expression patterns and the prognoses in terms of recurrence-free survival (RFS) and cancer-specific survival (CSS). A Kaplan–Meier analysis showed that patients with HE had a significantly worse RFS and CSS than those with LE (hazard ratio [HR]: 1.78, 95% confidence interval [CI]: 1.02–2.98, *p* = 0.027 and HR: 1.91, 95% CI: 1.08–3.38, *p* = 0.023, respectively). In a multivariate analysis, independent risk factors for worse RFS and CSS were shown as HE (HR: 1.96, 95% CI: 1.08–3.53, *p* = 0.026 and HR: 2.22, 95% CI: 1.14–4.31, *p* = 0.019, respectively) and lymph node metastasis (HR: 2.04, 95% CI: 1.09–3.84, *p* = 0.026 and HR: 1.19, 95% CI: 1.25–4.97, *p* = 0.009, respectively). The present study showed that AHNAK2 acts as a novel prognostic biomarker in patients with RC for BCa.

## 1. Introduction

Bladder cancer (BCa) is the most common malignancy of the urinary tract and the fourth most common cancer in men [1]. For the last three decades, radical cystectomy (RC) has been the gold-standard treatment in patients with muscle-invasive BCa (MIBC) and the non-muscle-invasive BCa (NMIBC) that is refractory to intravesical therapy. Despite advances in surgical techniques and an improved understanding of the role of pelvic lymphadenectomy, recurrences after RC usually occur within the first 2–3 years, giving a 5-year survival for only about 50% of patients [2,3]. Moreover, once MIBC metastasizes, a five-year survival rate was dismal at less than 10% even with salvage treatments [3].

To improve such unfavorable results following RC [4], great efforts have been made for the investigation of prognostic factors related to the surgery [5]. Currently, management of BCa still relies on histopathological parameters such as tumor stage, lymph node status, and lymphovascular invasion (LVI) [5]. Although these prognostic variables have been helpful in estimating the recurrence risk and survival outcomes of BCa, they do not largely play predictive roles in the individual strategy. For example, some studies have shown that the effect of neoadjuvant chemotherapy did not correlate with T stage [6,7]. Meanwhile, in several other cancers, the use of molecular biomarkers as a guide to personalized treatment has become standard, improving patients’ survival, especially in breast cancer [8]. Therefore, there is an urgent need to identify potential molecular markers of BCa, which can serve as not only prognostic values but act as potential therapeutic targets.

AHNAK2, also known as C14orf78, is a member of the AHNAK family and was originally found in mouse heart tissue extract, encoding a giant protein of more than 600 kilodaltons (kDa) [9]. Over the last seven years, overexpression of AHNAK2 has been reported to be associated with poor prognosis in clear cell renal cell carcinoma, pancreatic ductal adenocarcinoma, uveal melanoma, papillary thyroid carcinoma, and lung adenocarcinoma [10,11,12,13,14]. Furthermore, previous studies indicated AHNKA2 as a possible new therapeutic target in some cancers because it would play an important role in regulating multiple tumor progression pathways, including *mitogen*-*activated protein kinase (MAPK)*/*extracellular signal*-*regulated kinase (ERK)*, phosphatidylinositol 3-kinase (PI3K)/protein *kinase* B (AKT), hypoxia inducible factor-1α (HIF-1α), and transforming growth factor-β (TGF-β)/Smad3 [10,12,15,16]. However, there are no data from investigations of the expression levels of AHNAK2 in patients following RC that consider clinicopathological features.

We previously examined AHNAK in BCa tissues and successfully identified AHNAK2 in patients with RC for BCa [17]. The present study evaluates AHNAK2 expression levels in patients and retrospectively investigates an association between AHNAK2 expression levels and the prognosis adjusted by pathological variables obtained by RC.

## 2. Results

### 2.1. Tissues Immunostained for AHNAK2

Figure 1 shows representative tissue sections immunostained for AHNAK2 in normal urothelial and tumor tissues (200×). In non-neoplastic tissues, AHNAK2 was observed in the cytoplasm of smooth muscle cells in the muscular layer, peripheral nerve cells, endothelial cells, macrophages, and tumor stromal fibroblasts. No, or only a weak, expression was observed in the cytoplasm of normal urothelial cells (Figure 1A). In tumor tissues, AHNAK2 was variously observed in the cytoplasm and/or plasma and nuclear membrane of tumor cells (Figure 1B–D).

### 2.2. Patient Characteristics

This study included 97 (80.8%) men and 23 (19.2%) women. The median time to a follow-up appointment was 38.8 months (range: 0.7–283.3 months; mean: 69.3 months). The patients’ characteristics are listed in Table 1. Patients with low expression (LE) and high expression (HE) of AHNAK2 accounted for 46.7% (*n* = 56) and 53.3% (*n* = 64), respectively. Of all patients, 50.8% (*n* = 61) had tumor recurrence and 42.5% (*n* = 51) experienced cancer death, with a significantly higher proportion in the HE group than in the LE group in terms of both parameters (recurrence: 57.8% vs. 42.9%, respectively, *p* = 0.027; cancer death: 50.0% vs. 33.9%, respectively, *p* = 0.023).

Numbers of each clinical T stage before RC in the LE and the HE group were as follows; Ta: 1 (1.8%) and 1 (1.6%), Tis: 3 (5.4%) and 1 (1.6%), T1: 15 (26.8%) and 10 (15.6%), T2: 14 (25.0%) and 9 (14.1%), T3: 15 (26.8%) and 28 (43.7%), and T4: 8 (14.2%) and 15 (23.4%), respectively. In terms of clinical N stage, patients with node involvement were 5 in the LE and 11 in the HE group.

With the exclusion of a small number of unknown cases due to unavailable data, patients in the HE group had a significantly higher proportion of MIBC, lymph node metastasis, and LVI (*p* = 0.047, *p* = 0.027, and *p* = 0.003, respectively) than patients in the LE group.

Not all patients received adjuvant chemotherapy (AC), but all those who did a platinum-based chemotherapy. The recurrence rate in the HE group was not significantly different between patients with AC and those who without AC (n = 7: 58.3% and n = 30: 57.7%, respectively, *p* = 0.97); the results were similar in the LE group (n = 4: 50% and n = 20: 41.7%, respectively, *p* = 0.68). In terms of patients with AC (*n* = 20), there was no significant difference in the recurrence rate between the two groups (*p* = 0.71). Of all the patients who received salvage chemotherapy (SC) (*n* = 20), 80% (*n* = 16; n = 7: 35% in the LE group and n = 9: 45% in the HE group) received platinum-based chemotherapy for the disease progression after RC. The response rates of SC between the HE and LE groups were not significantly different (27.3% and 22.2%, respectively, *p* = 0.80), and 80% (4/5) of all patients with the response experienced cancer death thereafter.

### 2.3. Survival Analysis Using Kaplan-Meier Methods for RFS and CSS in Terms of Two Types of AHNAK2 Expression

A Kaplan–Meier analysis showed that patients in the HE group had a significantly worse recurrence-free survival (RFS) than those in the LE group (hazard ratio [HR]: 1.78, 95% confidence interval [CI]: 1.02–2.98, *p* = 0.027; Figure 2). The cumulative RFS rates for patients in the HE and LE groups were 62.1% and 85.2% at one year, 46.7% and 73.6% at two years, and 41.2% and 56.3% at five years, respectively. The mean times to recurrence after RC for patients in the HE and LE groups were 14.9 months and 25.2 months (*p* = 0.003), respectively. In terms of cancer death, a Kaplan–Meier analysis showed that patients in the HE group had a significantly worse cancer-specific survival (CSS) than those in the LE group (HR: 1.91, 95% CI: 1.08–3.38, *p* = 0.023; Figure 3). The cumulative CSS rates for patients in the HE and LE groups were 74.9% and 90.7% at one year, 52.3% and 80.0% at two years, and 46.7% and 67.7% at five years, respectively. The mean times to cancer death from RC in patients in the HE and LE groups were 15.4 months and 29.2 months (*p* = 0.006), respectively.

### 2.4. Univariate and Multivariate Analyses of Prognostic Factors for RFS and CSS

A univariate analysis showed that the recurrence was associated with HE, MIBC, lymph node metastasis, and LVI; a multivariate analysis adjusted for the effects of clinicopathological features showed that HE and lymph node metastasis were independent risk factors for the recurrence (Table 2). In terms of cancer death, a univariate analysis showed that cancer death was associated with HE, lymph node metastasis, and LVI; a multivariate analysis showed that HE and lymph node metastasis were independent risk factors for cancer death (Table 3).

### 2.5. Subgroup Analysis of Associations between AHNAK2 and Some Proteins

Table 4 shows associations in patients between the two types of AHNAK2 expressions and other proteins including S100A2, S100A4, S100A8, S100A9, and nestin. Patients in the HE group had a significantly higher proportion of S100A4, S100A8, S100A9, and nestin than those in the LE group.

## 3. Discussion

Currently, an evaluation of AHNAK2 in BCa has been very limited. As a diagnostic marker, AHNAK2 could immunohistochemically differentiate between inflammatory changes and carcinoma in situ (CIS) [18]. In terms of the prognostic value, only a few studies have been performed and reported that a higher expression of AHNAK2 was associated with shorter overall survival in BCa [13,19]. However, these previous studies were based on gene enrichment analyses and focused on an association between gene types and survival duration without adjusting clinical information. Hence, the present study retrospectively investigated the clinicopathological features and prognosis in 120 patients treated with RC in terms of the protein expression levels of AHNAK2. Consequently, we found that in patients with HE, the level of expression was related to biological aggressiveness such as for MIBC, lymph node metastasis, and LVI, with worse RFS and CSS and with about a year less time to recurrence and cancer death in comparison with those with LE. Furthermore, the multivariate analysis using the Cox proportional hazards regression model showed that HE and lymph node metastasis were independent predictors of worse RFS and CSS. Taken together, these results highlight the significant prognostic value of AHNAK2 in patients with BCa.

Pan-cancer analyses revealed the functional roles of AHNAK2 in the epithelial-mesenchymal transition (EMT), which plays a key role as a tumor promoter by allowing epithelial cells to gain a range of mesenchymal characteristics [13,14]. Analyses based on clinical specimens showed that AHNAK2 regulated the EMT via a TGF-β/Smad3 pathway in lung adenocarcinoma and a hypoxia inducible factor-1α/zinc finger E-box-binding homeobox 1 (HIF-1/ZEB1) pathway in clear cell renal cell carcinoma [10,16]. In terms of BCa, basic experiments with lung metastasis of BCa showed that such EMT activation via TGF-β/Smad3 and HIF-1α/ZEB1 was indeed shown to strongly contribute to the invasion and metastasis of BCa [20,21]. Because most of the S100 proteins have been considered as EMT facilitators in certain carcinoma cell lines [21], the present study showed that HE of AHNAK2 had a significantly higher proportion of S100A4, S100A8, and S100A9 expressions. Therefore, the EMT pathway potentially driven by TGF-β/Smad3 and HIF-1α/ZEB1 possibly explains how the high level of AHNAK2 may be an independent prognostic factor in the poor survival rates of patients treated with RC.

The poor prognostic value of AHNAK2 in BCa may also result from fibroblast growth factor-1 (FGF1) signaling. The Cancer Genome Atlas (TGCA) project recently indicated that the MAPK/ERK and PI3K/AKT pathways were potential BCa driver genes and that these downstream cascades were mainly activated by FGF receptors (FGFRs), especially FGFR1 and FGFR3 in BCa [22]. In particular, AHNAK2 is required for non-classical secretion of FGF1, which can universally activate all FGFRs [23,24]. Because the HE of AHNAK2 correlated with a significantly higher proportion of S100A8/9 and nestin in the current study, several in vitro studies indicated that S100A8/9 activated MAPK pathway in breast, colon, and prostate cancer, and nestin activated PI3K phosphorylation in glioblastoma and embryogenesis [25,26]. In fact, these potential relations between AHNAK2 and the MAPK/ERK and PI3K/AKT pathways were reported in lung adenocarcinoma and uveal melanoma [12,15]. Therefore, given the accelerated approval of FGFR inhibitors for patients with advanced BCa by the U.S. Food and Drug Administration, the antitumor effects of a combination of therapies targeting AHNAK2 and FGFRs might be worth analyzing in the future [22].

Additionally, we evaluated differences in clinical outcomes following AC and SC between the two AHNAK2 groups. In terms of AC, although the recurrence rate in the HE group was higher than that in the LE group, the difference did not reach statistical significance. Moreover, the response rate of SC in both groups was only about 25%, and almost all the responders experienced cancer death thereafter. The expression level of AHNAK2 in the RC specimen thus did not show predictive value in such cisplatin-based chemotherapy. However, emerging evidence suggests that immune evasion induced by EMT may largely contribute to cisplatin resistance, and the small sample size of patients with the postoperative chemotherapy in the present study may have insufficient power to draw any conclusion [27]. Notably, EMT may also be related to the resistance of immune checkpoint inhibitor (ICI) in some malignant cells, and a dataset of BCa from TCGA recently showed an association between a higher EMT-related gene expression and a lower response rate of nivolumab (the programmed death-1 inhibitor) [28,29]. Considering the possible oncogenic role of AHNAK2 via EMT described in the present study, further studies should be warranted to verify the predictive value of AHNAK2 for cisplatin and ICI in a large population in the future.

This study had some limitations. First, the study was based on the retrospective design with the small number of patients, which may have limited proper assessment of the correlation between expression level of AHNAK2 in the RC specimen and prognosis in patients with RC. Second, RC was performed by multiple surgeons, and the management of the postoperative chemotherapy such as the treatment intensity was decided by each doctor in charge, and these differences may have influenced our results. Third, in terms of the subgroup analysis, the times during which we investigated the expression levels of the five proteins in patients treated with RC varied. Subsequently, the total number of proteins investigated in each patient chronologically increased. Fourth, although we did not conduct an experiment to investigate the actual mechanism of AHNAK2, the results of our subgroup analysis may be helpful to infer the possible oncogenic role of AHNAK2 in BCa. Fifth, we did not include some patient characteristics—including smoking status—which potentially affect prognosis in BCa. However, we believe that a focus on pathological findings, when added to the AHNAK2 expression patterns, may simply allow us to explain the differences in the prognosis.

## 4. Materials and Methods

### 4.1. Patient Population

We retrospectively reviewed the clinical data of 161 consecutive patients with BCa who underwent RC with pelvic and iliac lymphadenectomy from 1990 to 2015 at Kitasato University Hospital, Japan. RC was performed for patients with non-MIBC that had been refractory to intravesical therapy and MIBC without distant metastasis. We excluded 10 patients who had histological variants of BCa, including squamous cell carcinoma, adenocarcinoma, and small cell carcinoma; 15 who had been previously treated with neoadjuvant chemotherapy; and 16 who were lost to follow-up. None of the remaining patients were treated preoperatively with either systemic chemotherapy or radiation therapy. The Ethics Committee of Kitasato University School of Medicine and Hospital approved the study (B17-010, B18-149). All participants were approached on the basis of approved ethical guidelines. The patients could refuse entry and discontinue participation at any time.

### 4.2. Patient Characteristics

The following data on patient characteristics were collected from the patients’ medical charts: Age at the RC; sex; pathological status including pT stage, pN stage, grade, LVI; CIS; history of AC; history of SC; recurrence; and mortality. BCa with ≥pT2 and ≤pT1 were also classified as MIBC and NMIBC, respectively. Tumor grade was assessed according to the 1973 World Health Organization grading system. The tumor stage was assessed according to the 2002 TNM classification of malignant tumors. LVI indicated the presence of cancer cells within the endothelial space. Cancer cells that merely invaded a vascular lumen were considered negative [30]. The chemotherapeutic response was evaluated by the Response Evaluation Criteria in Solid Tumors (RECIST) version 1.1 [31]. We categorized the patients as either responsive (complete response or partial response) or non-responsive (stable disease or disease progression).

### 4.3. Immunohistochemistry and Scoring

Formalin-fixed, paraffin-embedded tissue blocks representing the most invasive areas of each tumor were collected for further investigation. Normal urothelium was harvested from cystectomized specimens.

Three-micron-thick sections were immunostained using the BOND-MAX automated immunohistochemistry system and Bond Polymer Refine Detection kit DS 9800 (Leica Biosystems, Newcastle, UK), following our previous studies with minor modifications [32]; tissues obtained by RC were deparaffinized and pretreated with Bond Epitope Retrieval Solution 2 (Leica Biosystems) at 100 °C for 20 min. After washing and peroxidase blocking for 10 min, tissues were re-washed, and immunohistochemistry was performed for AHNAK2 on the specimens by rabbit polyclonal anti-AHNAK2 antibody (HPA004145; Sigma-Aldrich, St. Louis, MO, USA, diluted 1:1000). Sections were incubated with EnVision FLEX+ Rabbit Linker (Dako, Glostrup, Denmark) for 15 min. Finally, the sections were incubated with BOND Polymer (Leica Biosystems) for 10 min, developed with 3,3ʹ-diaminobenzidine (DAB) chromogen for 10 min, and counterstained with hematoxylin for 5 min. Sections treated with BOND Primary Antibody Diluent (Leica Biosystems) replacing the primary antibody were used as negative controls.

AHNAK2 was located in the cytoplasm and/or plasma and nuclear membrane of tumor cells, and the staining of tumor cells in these locations was considered positive. We scored the expression of AHNAK2 in the tumor cell using the following scheme: The intensity of staining was scored as 0 (negative), 1 (weak), 2 (moderate), or 3 (strong). The extent of staining was scored according to the percentage of positive tumor cells: 0 (none), 1 (1–25%), 2 (26–50%), 3 (51–75%), and 4 (76–100%). The two scores were multiplied, and the products (score) ranged from 0 to 12. A score between 0 and was considered as LE, whereas a score of 3 or more was considered as HE based on the median score of 3 [14]. All the immunostained sections were reviewed by two investigators (D.K. and Y.S.) without any knowledge of the clinical data. Discordant cases were reviewed and discussed until a consensus was reached.

### 4.4. Subgroup Analysis of Associations between AHNAK2 and Some Proteins

We additionally examined associations between AHNAK2 and some proteins, which we had previously reported in BCa, such as S100A2, S100A4, S100A8, S100A9, and nestin, for a better understanding of the mechanism of AHNAK2. The expressions of these proteins were calculated by a sum index score and categorized as previously described; S100 families were categorized as normal or abnormal and nestin as negative or positive [33,34,35].

### 4.5. Statistical Analysis

Comparisons of clinicopathological futures between the LE and HE of AHNAK2 were performed using the chi-square test (or Fisher’s exact test, if appropriate) for categorical variables, and using the Mann–Whitney U test for continuous variables. RFS and CSS were estimated by the Kaplan–Meier method with the log-rank test. Multivariate analyses were performed with the Cox proportional hazards regression model, controlling for the effects of clinicopathological parameters. All statistical analyses were performed with Stata ver. 14 for Windows (Stata, Chicago, IL, USA). All *p* values were two-sided, and *p* < 0.05 was considered statistically significant.

## 5. Conclusions

The present study found that high expression levels of AHNAK2 were associated with aggressive pathological findings obtained by RC and were an independent predictor of worse RFS and CSS in patients with RC. Hence, we believe that AHNAK2 may act as a novel prognostic biomarker in the patients. Furthermore, associations between HE and S100A4, S100A8, S100A9, and nestin may highlight AHNAK2 as a novel therapeutic target of BCa. Further studies are warranted to elucidate the reported complex mechanisms of AHNAK2 in BCa.

## Figures and Tables

**Figure 1 cancers-13-01748-f001:**
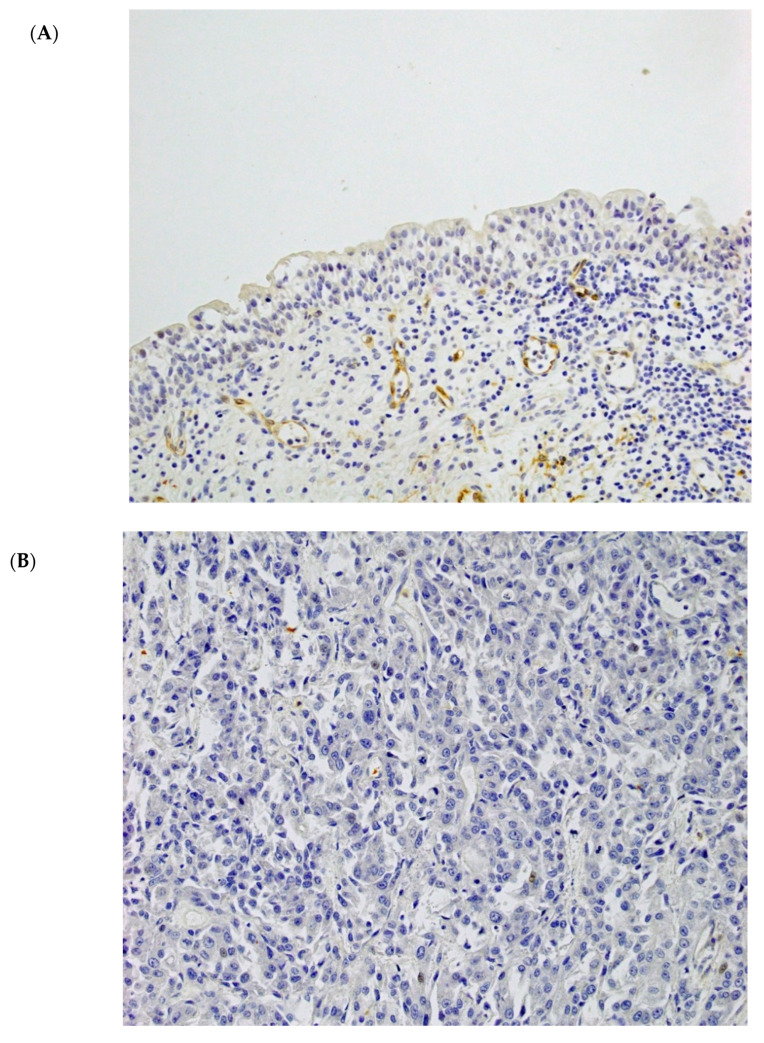
Immunohistochemical analysis of AHNAK2 expression in normal urothelial and bladder carcinoma tissues. Microscopic images are representative normal urothelial and bladder cancer (BCa) tissues for AHNAK2 staining (200× magnification). (**A**) AHNAK2-negative normal urothelial tissues. (**B**) AHNAK2-negative BCa tissues: score 0. (**C**) AHNAK2-negative BCa tissues: Score 2. (**D**) AHNAK2-positive BCa tissues: Score 12.

**Figure 2 cancers-13-01748-f002:**
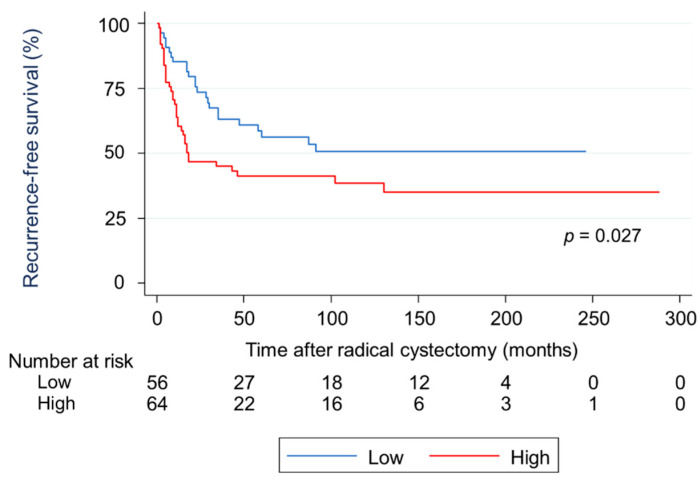
Survival analysis using Kaplan–Meier methods for recurrence-free survival in terms of two types of AHNAK2 expression. Patients in the high expression (HE) group showed a significantly worse recurrence-free survival than those in the low expression (LE) group.

**Figure 3 cancers-13-01748-f003:**
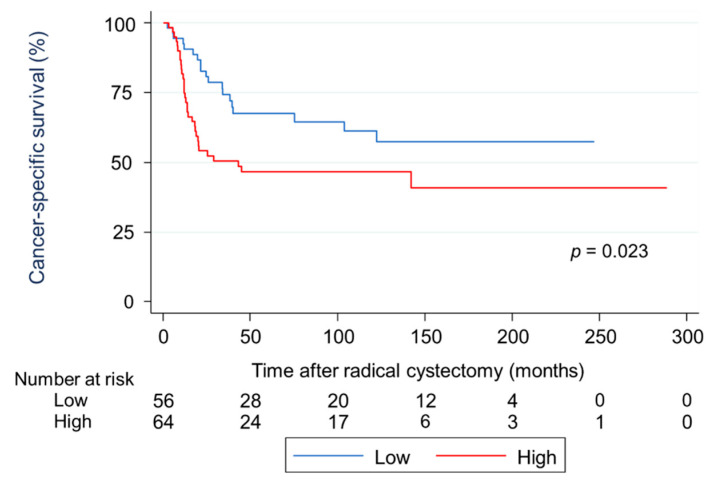
Survival analysis using Kaplan–Meier methods for cancer-specific survival in terms of two types of AHNAK2 expression. Patients in the high expression (HE) group showed a significantly worse cancer-specific survival than those in the low expression (LE) group.

**Table 1 cancers-13-01748-t001:** Comparison of clinical and pathological characteristics of patients with either low or high expressions of AHNAK2.

	LE (*n* = 56)	HE (*n* = 64)	*p*-Value
Age			
≤65	29 (51.8)	33 (51.6)	0.98
>65	27 (48.2)	31 (48.4)	
Sex			
Male	51 (91.1)	46 (71.2)	0.01
Female	5 (8.9)	18 (28.8)	
T stage			
pTa	2 (3.6)	0	0.047
pTis	2 (3.6)	1 (1.6)	
pT1	11 (19.6)	7 (10.9)	
pT2	18 (32.1)	11 (17.2)	
pT3	16 (28.6)	30 (46.9)	
pT4	7 (12.5)	15 (23.4)	
N stage			
pN0	45 (80.3)	43 (67.2)	0.027
≥pN1	10 (17.8)	16 (25.0)	
Unknown	1 (1.8)	5 (7.8)	
Grade			
G1/2	23 (41.1)	23 (35.9)	0.57
G3	32 (57.1)	41 (64.1)	
Unknown	1 (1.8)	0	
LVI			
Negative	24 (42.9)	15 (23.4)	0.003
Positive	28 (50.0)	43 (67.2)	
Unknown	4 (7.1)	6 (9.4)	
CIS			
Negative	49 (87.5)	55 (85.9)	0.97
Positive	7 (12.5)	8 (12.5)	
Unknown	0	1 (1.6)	
Adjuvant chemotherapy			
Yes	8 (14.3)	12 (18.8)	0.51
No	48 (85.7)	52 (81.2)	
Salvage chemotherapy			
Response	2 (22.2)	3 (27.3)	0.80
No Response	7 (77.8)	8 (72.7)	
Recurrence			
Yes	24 (42.9)	37 (57.8)	0.027
No	32 (57.1)	27 (42.2)	
Cancer death			
Yes	19 (33.9)	32 (50.0)	0.023
No	37 (66.1)	32 (50.0)	
Follow-up, months (IQR)	51.0 (21–133)	20.0 (11–98.5)	0.075

Unless otherwise stated, values are medians with ranges in parentheses or numbers of patients with percentages in parentheses. LE, low expression; HE, high expression; LVI, lymphovascular invasion; CIS, carcinoma in situ; IQR, interquartile range.

**Table 2 cancers-13-01748-t002:** Univariate and multivariate analyses for worse recurrence-free survival. AHNAK2, T stage, N stage, grade, lymphovascular invasion (LVI), and carcinoma in situ (CIS) were evaluated, and statistically significant values are highlighted in bold.

Variable	Category	Univariate Analysis			Multivariate Analysis		
		HR	95% CI	*p*-Value	HR	95% CI	*p*-Value
AHNAK2	HE	1.78	1.02–2.98	0.027	1.96	1.08–3.53	0.026
	LE	1.0			1.0		
T stage	MIBC	2.24	1.14–4.41	0.019	1.66	0.64–4.28	0.30
	NMIBC	1.0			1.0		
N stage	≥pN1	2.86	1.67–4.89	<0.001	2.04	1.09–3.84	0.026
	pN0	1.0			1.0		
Grade	G3	1.47	0.83–2.60	0.18	1.12	0.55–2.27	0.75
	G1/2	1.0			1.0		
LVI	Positive	2.52	1.35–4.71	0.004	1.12	0.55–2.27	0.75
	Negative	1.0			1.0		
CIS	Positive	0.64	0.29–1.41	0.27	1.16	0.48–2.84	0.73
	Negative	1.0			1.0		

HR, hazard ratio; CI, confidence interval; HE, high expression; LE, low expression; MIBC, muscle-invasive bladder cancer; NMIBC, nonmuscle-invasive bladder cancer; LVI, lymphovascular invasion; CIS, carcinoma in situ.

**Table 3 cancers-13-01748-t003:** Univariate and multivariate analyses for worse cancer-specific survival. AHNAK2, T stage, N stage, grade, LVI, and CIS were evaluated, and statistically significant values are highlighted in bold.

Variable	Category	Univariate Analysis			Multivariate Analysis		
		HR	95% CI	*p*-Value	HR	95% CI	*p*-Value
AHNAK2	HE	1.91	1.08–3.38	0.023	2.22	1.14–4.31	0.019
	LE	1.0			1.0		
T stage	MIBC	2.01	0.98–4.13	0.057	1.44	0.50–4.11	0.68
	NMIBC	1.0			1.0		
N stage	≥pN1	3.03	1.69–5.45	<0.001	1.19	1.25–4.97	0.009
	pN0	1.0			1.0		
Grade	G3	1.65	0.87–3.15	0.13	1.17	0.54–2.56	0.69
	G1/2	1.0			1.0		
LVI	Positive	2.43	1.23–4.80	0.01	1.19	0.53–2.64	0.68
	Negative	1.0			1.0		
CIS	Positive	0.42	0.15–1.18	0.101	0.57	0.17–1.91	0.37
	Negative	1.0			1.0		

HR, hazard ratio; CI, confidence interval; HE, high expression; LE, low expression; MIBC, muscle-invasive bladder cancer; NMIBC, nonmuscle-invasive bladder cancer; LVI, lymphovascular invasion; CIS, carcinoma in situ.

**Table 4 cancers-13-01748-t004:** Associations in patients between the two types of AHNAK2 expressions and S100A2, S100A4, S100A8, S100A9, and nestin. Statistically significant values are highlighted in bold.

Protein	LE (*n* = 56)	HE (*n* = 64)	*p*-Value
S100A2			
Normal	16 (28.6)	12 (18.8)	0.15
Abnormal	19 (33.9)	31 (48.4)	
S100A4			
Normal	22 (39.3)	15 (23.4)	0.022
Abnormal	13 (23.2)	28 (43.8)	
S100A8			
Normal	24 (42.9)	20 (31.1)	0.015
Abnormal	7 (12.5)	22 (34.4)	
S100A9			
Normal	28 (50)	20 (31.1)	<0.001
Abnormal	3 (5.4)	22 (34.4)	
Nestin			
Negative	36 (64.3)	35 (54.6)	0.005
Positive	1 (1.8)	12 (18.8)	

Values are numbers of patients with percentages in parentheses. The numbers (percentages) of patients in the LE and HE groups for whom protein information was not available are as follows: S100A2/4: 21 (37.5) and 21 (32.8), respectively; S100A8/9: 25 (44.6) and 22 (34.4), respectively; Nestin, 19 (33.9) and 17 (26.6), respectively. LE, low expression; HE, high expression.

## Data Availability

The datasets used and/or analyzed during the current study are available from the corresponding author on reasonable request.

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
