# Peer review of "Prognostic Impact of AHNAK2 Expression in Patients Treated with Radical Cystectomy"

_cancers, 2021, doi:10.3390/cancers13081748_

Round 1
Reviewer 1 Report
This study has analyzed the prognostic impact of the expression of AHNAK2 protein in patients with bladder cancer (BC) treated with radical cystectomy (RC). The results suggest the potential impact of AHNAK2 expression as a prognostic biomarker in these patients. The role of AHNAK2 has rarely been analyzed in urothelial cancer, and the results can therefore be considered relatively novel. The text is well written and the results are clearly presented. However, the study has some limitations and some modifications are required.
- This is a retrospective study and has included a relatively small number of patients (120), which limits the possibility of obtaining definite conclusions. The authors should comment on this in their discussion.
- Currently, the standard treatment for muscle-invasive bladder cancer (MIBC) is neoadjuvant chemotherapy followed by cystectomy. This limits the applicability of this study’s results in clinical practice.
- Some additional information should be reported about the characteristics of patients included in this study:
- How were the patients selected for this analysis? Were patients with non-MIBC included? The clinical stage of the patients before cystectomy should be provided.
- Were the patients with MIBC included in the study not considered candidates for neoadjuvant chemotherapy?
- The pathological stage (at cystectomy) of the patients should be explained in more detail according to the TNM classification.
- Some of the comments in the Discussion about the potential role of AHNAK2 as a predictive biomarker of response to immunotherapy or targeted therapy (eg, FGFRinh) cannot be inferred from the results of the study, although they merit being examined in future. Therefore, the authors should tone down these comments and focus more on conclusions that can be drawn from the results of the study.
Author Response
RE: Mn.No. cancers-1154956Title: " Prognostic impact of AHNAK2 expression in patients treated with radical cystectomy"
Thank you for the critical review of our manuscript and the opportunity to submit revisions. We have reviewed the comments from the Reviewer 1, and made revisions where we feel appropriate. A list of the response to each of the comments as follows.
Reviewer: 1
#1. This is a retrospective study and has included a relatively small number of patients (120), which limits the possibility of obtaining definite conclusions. The authors should comment on this in their discussion.
Answer: Thank you so much for your useful comment. We added sentences you mentioned as the main limitation as below.
Page13, “Discussion” paragraph5, line 1-3.
First, the study was based on the retrospective design with the small number of patients, which may have limited proper assessment of the correlation between expression level of AHNAK2 in the RC specimen and prognosis in patients with RC.
#2. Currently, the standard treatment for muscle-invasive bladder cancer (MIBC) is neoadjuvant chemotherapy followed by cystectomy. This limits the applicability of this study’s results in clinical practice.
Answer: We appreciate reviewer’s comments on neoadjuvant chemotherapy (NAC). As you mentioned, evidence favors NAC over adjuvant chemotherapy in radical cystectomy (RC), reporting an overall survival benefit of 8% at 5 years after RC [1]. However, unfortunately, the present cohort had only 15 patients with NAC during the study periods. Therefore, we did not include such patients due to the limited statistical power. As a result, the data we obtained by RC could be shown as relatively pure tumor characteristics because the specimens were not affected by NAC. Therefore, we believe that our study design without patients with NAC, or rather with a focus on pathological findings obtained by RC is also valuable, simply allowing us to explain the differences in the prognosis among the two AHNAK2 expression groups.
Additionally, reasons for the small number of patients with NAC in the present study may be as follows; Despite level 1 evidence of NAC in patients with RC [1], compliance with this guideline recommendation is underutilized worldwide, for example, reporting about 20% at best in some series [2]. Literatures find that such low prevalence of NAC may arise from patients’ comorbidities, lower performance status, poor renal function and the concern of delay of RC [2]. Furthermore, many physicians feel that pT2 tumors can be adequately treated with RC alone because these patients have a 5-year survival of about 80% when treated with RC alone [3]. Finally, of great importance is that patients who underwent RC until the end of 2003 (published year of the reference number 1 in this document) in the present study accounted for about 60% of all (71/120: 59.2%); The evidence of NAC had not been so confirmed at that period as now. Taken these together, we believe that the small number of patients with NAC during the study period is acceptable as the real-world data.
#3. Some additional information should be reported about the characteristics of patients included in this study:
1) How were the patients selected for this analysis? Were patients with non-MIBC included? The clinical stage of the patients before cystectomy should be provided.
Answer: Thank you very much for your sharp opinion. In our institution, Kitasato University Hospital, racial cystectomy (RC) is performed for patients with non-MIBC that have been refractory to intravesical therapy and MIBC without distant metastasis. Subsequently, we simply reviewed all patients who underwent RC from 1990 to 2015 at Kitasato University Hospital (n=161) by a retrospective manner. Finally, 120 patients were eligible for the present study, with exclusions of 10 patients with histological variants of BCa, 15 with neoadjuvant chemotherapy and 16 without follow-up. In other words, we did not select patients for a particular purpose. However, the surgical indication of RC in the present study was ambiguous, and we added the indication in the method section and the clinical stage in the result section as below.
Page14, “Material and Methods, Patient Population” paragraph1, line 3-4.
RC was performed for patients with non-MIBC that had been refractory to intravesical therapy and MIBC without distant metastasis.
Page6, “Results, Patient Characteristics” paragraph2, line 1-4.
Numbers of each clinical T stage before RC in the LE and the HE group were as follows; Ta: 1 (1.8%) and 1 (1.6%), Tis: 3 (5.4%) and 1 (1.6%), T1: 15 (26.8%) and 10 (15.6%), T2: 14 (25.0%) and 9 (14.1%), T3: 15 (26.8%) and 28 (43.7%) and T4: 8 (14.2%) and 15 (23.4%), respectively. In terms of clinical N stage, patients with node involvement were 5 in the LE and 11 in the HE group.
2) Were the patients with MIBC included in the study not considered candidates for neoadjuvant chemotherapy?
Answer: Thank you very much for your interests. First of all, the patient selection for the present study was not based on whether those who were eligible or ineligible for neoadjuvant chemotherapy (NAC). So, some patients with MIBC would have been candidates for NAC and others would not have been. As we answered Reviewer 1 question #2, in the present study, a large population of patients had radical cystectomy from 1990 to the early 2000s. Therefore, the evidence of NAC had not been so confirmed at that period as now, and there were only 15 patients with NAC in the present study. We believe that such small number of patients with NAC does not achieve appropriate statistical power. However, your comment is very valuable, and we will accumulate the patients and perform these analyses in another study which will have your concept. Thank you for your understanding.
3) The pathological stage (at cystectomy) of the patients should be explained in more detail according to the TNM classification.
Answer: Thank you so much for the advice. We rewrote the pathological stage in Table 1 according to your advice.
Page7, “Table 1” T stage.
#4. Some of the comments in the Discussion about the potential role of AHNAK2 as a predictive biomarker of response to immunotherapy or targeted therapy (eg, FGFRinh) cannot be inferred from the results of the study, although they merit being examined in future. Therefore, the authors should tone down these comments and focus more on conclusions that can be drawn from the results of the study.
Answer: Thank you very much for your useful comment. We totally agree to your opinion. We added the statistical data of the recurrence rate in the result section, and corrected paragraph 4 in the discussion section. Then we removed the sentences related to predictive role of AHNAK2 for ICI and FGFRI form the conclusion section.
Page6, “Results, Patient Characteristics” paragraph4, line 4-6.
In terms of patients with AC (n = 20), there was no significant difference in the recurrence rate between the two groups (P = 0.71).
Page13, “Discussion” paragraph4, line 1-14.
Additionally, we evaluated differences in clinical outcomes following AC and SC between the two AHNAK2 groups. In terms of AC, although the recurrence rate in the HE group was higher than that in the LE group, the difference did not reached a statistical significance. Moreover, the response rate of SC in both groups was only about 25%, and almost all the responders experienced cancer death thereafter. Expression level of AHNAK2 in the RC specimen thus did not show predictive value in such cisplatin-based chemotherapy. However, emerging evidence suggest that immune evasion induced by EMT may largely contribute to cisplatin resistance, and the small sample size of patients with the postoperative chemotherapy in the present study may have insufficient power to draw any conclusion [27]. Notably, EMT may also be related to the resistance of immune checkpoint inhibitor (ICI) in some malignant cells, and a dataset of BCa from TCGA recently showed an association between a higher EMT-related gene expression and a lower response rate of nivolumab (the programmed death-1 inhibitor) [28, 29]. Considering the possible oncogenic role of AHNAK2 via EMT described in the present study, further studies should be warranted to verify the predictive value of AHNAK2 for cisplatin and ICI in a large population in the future.
Page16, “Conclusions” paragraph1, line 1-6.
The present study found that high expression levels of AHNAK2 were associated with aggressive pathological findings obtained by RC and were an independent predictor of worse RFS and CSS in patients with RC. Hence, we believe that AHNAK2 may act as a novel prognostic biomarker in the patients. Furthermore, associations between HE and S100A4, S100A8, S100A9, and nestin may highlight AHNAK2 as a novel therapeutic target of BCa. Further studies are warranted to elucidate the reported complex mechanisms of AHNAK2 in BCa.
Reference
- Stein, J.P. Editorial: Contemporary Concepts of Radical Cystectomy And The Treatment of Bladder Cancer. J Urology2003, 169, 116–117, doi:10.1097/00005392-200301000-00028.
- Zamboni, S.; Moschini, M.; Antonelli, A.; Simeone, C.; Belotti, S.; Cristinelli, L.; Montorsi, F.; Briganti, A.; Gallina, A.; Salonia, A.; et al. How to Improve Patient Selection for Neoadjuvant Chemotherapy in Bladder Cancer Patients Candidate for Radical Cystectomy and Pelvic Lymph Node Dissection. World J Urol2020, 38, 1229–1233, doi:10.1007/s00345-019-02916-2.
- Shariat, S.F.; Karakiewicz, P.I.; Palapattu, G.S.; Lotan, Y.; Rogers, C.G.; Amiel, G.E.; Vazina, A.; Gupta, A.; Bastian, P.J.; Sagalowsky, A.I.; et al. Outcomes of Radical Cystectomy for Transitional Cell Carcinoma of the Bladder: A Contemporary Series From the Bladder Cancer Research Consortium. J Urology2006, 176, 2414–2422, doi:10.1016/j.juro.2006.08.004.
Reviewer 2 Report
This study describe the use of IHC of AHNAK2 among cancer tissue of radical cystectomy patients. The levels of AHNAK2 were categorized to high and low level (HE vs LE). Patients with HL of AHNAK2 were associated with advanced disease, cancer progression and death of disease. By multivariate analysis, HE of AHNAK2 and LN+ was associated with cancer death.
AHNAK2 was not associated with response to salvage therapy in patients with relapse following cystectomy. In addition, response to neo-adjuvant therapy could not estimate since no patients treated with NAC were included in the study.
The major limitation is the lack of significance for therapy according to AHNAK2 expression.
The study is an interesting one and present new data of AHNAK2 despite its limitations (retrospective, no NAC patients, lack of lymph node numbers as a predictor, differences in IHC methods).
Minor corrections should be made as:
The two scores were multiplied and the products (score) ranged from 0 to 12. A score between 0 and was considered as LE, whereas a score of 3 or more was considered as HE based on the median score of 3 [14]. All the immunostained sections were reviewed by two investigators (D.K. and Y.S.) with-out any knowledge of the clinical data. Discordant cases were reviewed and discussed until a consensus was reached.
Please add references regarding the expression of AHNAK3 and bladder cancer such as:
Clinical Trial Am J Pathol . 2019 Mar;189(3):619-631. doi: 10.1016/j.ajpath.2018.11.018. Epub 2019 Feb 12.Integrated Fourier Transform Infrared Imaging and Proteomics for Identification of a Candidate Histochemical Biomarker in Bladder Cancer
Affiliations expand- PMID: 30770125
- DOI: 10.1016/j.ajpath.2018.11.018
Author Response
RE: Mn.Title: " Prognostic impact of AHNAK2 expression in patients treated with radical cystectomy"
Thank you for the critical review of our manuscript and the opportunity to submit revisions. We have reviewed the comments from the reviewer 2 and made revisions where we feel appropriate. A list of the response to each of the comments as follows.
Reviewer 2
Please add references regarding the expression of AHNAK2 and bladder cancer such as: Clinical Trial Am J Pathol. 2019 Mar;189(3):619-631. doi: 10.1016/j.ajpath.2018.11.018. Epub 2019 Feb 12.
Answer: Thank you for your kindness. We had already quoted the paper, which you recommend, as the reference number 18 in the manuscript.
Reviewer 3 Report
Review Report
In the manuscript entitled- “Prognostic impact of AHNAK2 expression in patients treated with radical cystectomy” the authors studied associations between the high and low AHNAK2 expression patterns in patients undergoing radical cystectomy for bladder cancer with prognosis in terms of recurrence-free survival and cancer-specific survival. The results showed that high expression of AHNAK2 is associated with worse recurrence-free and cancer-specific survival than those with low expression.
Major Comments-
Although there are just few papers addressing this subject the manuscript is based on exploratory analysis and lacks data supporting the oncogenic role of AHKNA2 in bladder cancer.
The authors should try to minimize the heterogeneity in patients in terms of treatment and risk factors.
The discussion section is long, indefinite and largely based on speculation.
Author Response
RE: Mn.Title: " Prognostic impact of AHNAK2 expression in patients treated with radical cystectomy"
Thank you for the critical review of our manuscript and the opportunity to submit revisions. We have reviewed the comments from the reviewer 3 and made revisions where we feel appropriate. A list of the response to each of the comments as follows.
Reviewer 3
#1. The authors should try to minimize the heterogeneity in patients in terms of treatment and risk factors.
Answer: We really appreciate your suggestion on this point. However, we are very afraid that we had tried to make relatively homogenous cohort in the present study. In terms of treatment risk, not all patients received adjuvant chemotherapy (AC), but all those who did a platinum-based chemotherapy. Furthermore, with respect to the treatment effect, AC did not have a significant impact on the recurrence rate in the two AHNAK2 expression groups; The recurrence rate in the HE group was not significantly different between patients with AC and those who without AC (58.3% [7/12] and 57.7% [30/52], respectively, P = 0.97); The results were similar in the LE group (50% [4/8] and 41.7% [20/48], respectively, P = 0.68). These unfavorable treatment results were also found in patients with salvage chemotherapy (SC); Of all the patients who received SC (n = 20), 80% (n = 16; LE: 35% [n = 7], HE: 45% [n = 9]) received platinum-based chemotherapy for the disease progression after RC. Then, the response rate of SC was approximately 25% in the two expression groups, and 80% [4/5] of all patients with treatment response experienced cancer death thereafter. Basically, although cisplatin-based chemotherapy has well known antitumor effect against urothelial cancer, there has been no level l evidence of AC in contrast to neoadjuvant chemotherapy, and the evidence level of SC have been more dismal [4]. Therefore, in the present study, we believe that relatively fair comparisons were performed in terms of the postoperative treatment factors. However, there remains a question related to what you pointed out because the treatment dose and schedule were decided by each doctor in charge. Then, we corrected the limitation section as below.
Second, we would like to discuss about the patient characteristics in the present study. As you pointed out, for example, smoking cessation after the diagnosis of bladder cancer would favorably affect clinical outcomes [5, 6]. Moreover, some factors have been also hypothesized to be associated with worse prognosis, such as occupational, lifestyle and nutritional modifiable factors. Therefore, we did not include some patient factors in order to make mainly focus on pathological findings, allowing us to simply explain the differences in prognosis between patients with high and low expression groups. Furthermore, we would like to demonstrate the association of AHNAK2 expressions with clinicopathological findings. In the present study, some factors showed statistical significances between the LE and the HE group, which we do not think as heterogeneity. In order to compensate these factors, we analyzed multivariate analyses for prognosis. These results revealed AHNAK2 was an independent risk factor in our cohort. However, what you suggested was clinically important, and we had already referred to this point in the limitation section in the manuscript as below. Thank you for your valuable comments and understanding.
Page13, “Discussion” paragraph5, line 4-6.
Second, RC was performed by multiple surgeons, and the management of the postoperative chemotherapy such as the treatment intensity was decided by each doctor in charge, and these differences may have influenced our results.
Page14, “Discussion” paragraph5, line 10-13.
Fifth, we did not include some patient characteristics—including smoking status—which potentially affect prognosis in BCa. However, we believe that a focus on pathological findings, when added to the AHNAK2 expression patterns, may simply allow us to explain the differences in the prognosis.
#2. The discussion section is long, indefinite and largely based on speculation
Answer: Thank you so much for the useful advice. We totally agree to your comment especially paragraph 4 in the discussion section. We corrected and shortened it from 266 to 209 words and removed the sentences related to predictive role of AHNAK2 for ICI and FGFRI form the conclusion section accordingly.
Page13, “Discussion” paragraph4, line 1-15.
Additionally, we evaluated differences in clinical outcomes following AC and SC between the two AHNAK2 groups. In terms of AC, although the recurrence rate in the HE group was higher than that in the LE group, the difference did not reached a statistical significance. Moreover, the response rate of SC in both groups was only about 25%, and almost all the responders experienced cancer death thereafter. Expression level of AHNAK2 in the RC specimen thus did not show predictive value in such cisplatin-based chemotherapy. However, emerging evidence suggest that immune evasion induced by EMT may largely contribute to cisplatin resistance, and the small sample size of patients with the postoperative chemotherapy in the present study may have insufficient power to draw any conclusion [27]. Notably, EMT may also be related to the resistance of immune checkpoint inhibitor (ICI) in some malignant cells, and a dataset of BCa from TCGA recently showed an association between a higher EMT-related gene expression and a lower response rate of nivolumab (the programmed death-1 inhibitor) [28, 29]. Considering the possible oncogenic role of AHNAK2 via EMT described in the present study, further studies should be warranted to verify the predictive value of AHNAK2 for cisplatin and ICI in a large population in the future.
Page16, “Conclusions” paragraph1, line 1-6.
The present study found that high expression levels of AHNAK2 were associated with aggressive pathological findings obtained by RC and were an independent predictor of worse RFS and CSS in patients with RC. Hence, we believe that AHNAK2 may act as a novel prognostic biomarker in the patients. Furthermore, associations between HE and S100A4, S100A8, S100A9, and nestin may highlight AHNAK2 as a novel therapeutic target of BCa. Further studies are warranted to elucidate the reported complex mechanisms of AHNAK2 in BCa.
Reference
4. Pak, S.; You, D.; Jeong, I.G.; Song, C.; Lee, J.-L.; Hong, B.; Hong, J.H.; Kim, C.-S.; Ahn, H. Adjuvant Chemotherapy versus Observation after Radical Cystectomy in Patients with Node-Positive Bladder Cancer. Sci Rep-uk2019, 9, 8305, doi:10.1038/s41598-019-44504-9.
5. Crivelli, J.J.; Xylinas, E.; Kluth, L.A.; Rieken, M.; Rink, M.; Shariat, S.F. Effect of Smoking on Outcomes of Urothelial Carcinoma: A Systematic Review of the Literature. Eur Urol2014, 65, 742–754, doi:10.1016/j.eururo.2013.06.010.
6. Westhoff, E.; Witjes, J.A.; Fleshner, N.E.; Lerner, S.P.; Shariat, S.F.; Steineck, G.; Kampman, E.; Kiemeney, L.A.; Vrieling, A. Body Mass Index, Diet-Related Factors, and Bladder Cancer Prognosis: A Systematic Review and Meta- Analysis. Bladder Cancer 2018, 4, 91–112, doi:10.3233/blc-170147.
Round 2
Reviewer 3 Report
The authors have revised the manuscript considerably. The hypothesis, the methods, the statistical analysis and conclusions now look appropriate. The authors have also provided a rationale behind the heterogeneity in this study and how it reflects the real clinical scenario. The authors have also shortened the discussion section and it looks relevant to the data shown in the manuscript. I would now recommend the revised version of this manuscript suitable for publication in this journal.